# Association between obesity and mortality from hematological malignancies among Japanese adults: The Japan Collaborative Cohort study

Hana Wakasa[1], Satoshi Sunohara[1,2], Takashi Kimura[3], Takaya Ichikawa[4], Akiko Tamakoshi[3]*

1 Department of Public Health, Graduate School of Medicine, Hokkaido University, Sapporo, Japan,
2 Clinical Training Center, Hokkaido University Hospital, Sapporo, Japan, 3 Department of Public Health, Faculty of Medicine, Hokkaido University, Sapporo, Japan, 4 Department of Hematology, Faculty of Medicine, Hokkaido University, Sapporo, Japan

* tamaa@pop.med.hokudai.ac.jp

## Abstract

Obesity is a growing global health concern and has been associated with increased mortality from various cancer types, including hematological malignancies. However, evidence for this association in Asian populations, particularly among Japanese adults, remains limited. Thus, this study aimed to examine the association between obesity and mortality due to hematological malignancies. Data from 97,073 participants in the Japan Collaborative Cohort (JACC) Study were analyzed. The participants were followed for a mean duration of 17 years. Body mass index (BMI) was calculated using self-reported height and weight and categorized as underweight ($<18.5\,kg/m^2$), normal-weight ($18.5$–$24.9\,kg/m^2$), overweight ($25.0$–$29.9\,kg/m^2$), and obesity ($\geq30.0\,kg/m^2$). Mortality data for hematological malignancies were obtained from death certificates. Cause-specific hazard ratios (HRs) and 95% confidence intervals (CIs) were estimated using Cox proportional hazards models, with adjustments made for demographic, lifestyle, and socioeconomic factors. During follow-up, 479 died from hematological malignancies, including lymphoma (n = 200), multiple myeloma (n = 107), and leukemia (n = 166; 106 myeloid leukemia). Compared with normal-weight individuals, those classified as obese exhibited a significantly higher risk of mortality from all hematological malignancies (HR: 1.78; 95% CI: 1.02–3.11), multiple myeloma (HR: 2.75; 95% CI: 1.09–6.94), leukemia (HR: 2.47; 95% CI: 1.07–5.69), and particularly myeloid leukemia (HR: 3.89; 95% CI: 1.66–9.11). No significant association was observed between BMI and lymphoma-related mortality. Obesity is significantly associated with increased mortality from multiple myeloma and leukemia, especially myeloid leukemia, in Japanese adults. These findings underscore the importance of obesity as a modifiable risk factor for certain hematological malignancies in this population.

**Data availability statement:** Data cannot be shared publicly because of the data used in this study contain potentially sensitive information. The data provider, the Ministry of Health, Labour and Welfare of Japan, has imposed restrictions that prevent the data from being made publicly available. Requests for access to the data may be directed to the JACC Study Group, which serves as the non-author contact point for data access inquiries. Contact information: JACC Study Group JACC_study@med.hokudai.ac.jp.

**Funding:** AT received funding from Grants-in-Aid for Scientific Research from the Ministry of Education, Culture, Sports, Science and Technology of Japan (MEXT) (Monbusho); Grants-in-Aid for Scientific Research on Priority Areas of Cancer; Grants-in-Aid for Scientific Research on Priority Areas of Cancer Epidemiology from MEXT (MonbuKagaku-sho) (Nos. 61010076, 62010074, 63010074, 1010068, 2151065, 3151064, 4151063, 5151069, 6279102, 11181101, 17015022, 18014011, 20014026, 20390156, and 26293138); Japan Society for the Promotion of Science, JSPS KAKENHI Grant Number JP 16H06277 (CoBiA) , and 22H04923; grant–in–aid from the Ministry of Health, Labor and Welfare, Health and Labor Sciences research grants, Japan (Research on Health Services: H17–Kenkou–007; Comprehensive Research on Cardiovascular Disease and Life–Related Disease: H18–Junkankitou [Seishuu]–Ippan–012; H19–Junkankitou [Seishuu]–Ippan–012; H20–Junkankitou [Seishuu]–Ippan–013; H23–Junkankitou [Seishuu]–Ippan–005; H26-Junkankitou [Seisaku]-Ippan-001; H29–Junkankitou–Ippan–003; 20FA1002; 23FA1006); the National Cancer Center Research and Development Fund (27-A-4, 30-A-15, 2021-A-16, 2024-A-14).

**Competing interests:** The authors have declared that no competing interests exist.

## Introduction

Obesity is a significant global health concern. In 2021, the estimated number of individuals classified as overweight or obese reached approximately 1.00 billion for men and 1.11 billion for women [1]. These figures represent increases of 155.1% for men and 104.9% for women compared with those reported in 1990. In addition, by 2050, more than half of the global adult population is expected to become overweight or obese [1]. Thus, the global burden of obesity has persisted over time. In Japan, the prevalence of obesity in 2023 was reported at 31.5% among men and 21.1% among women [2], with projections indicating further increases by 2050 [1].

Extensive evidence has identified obesity as a significant risk factor for various diseases and health conditions, including cardiovascular disease [3–5], type 2 diabetes [4,6,7], mental health problems [8–10], and several forms of cancer [11–13].

Cancer represents a major global public health concern. Approximately 20 million new cases and 9.7 million deaths have been reported annually worldwide [14]. The global macroeconomic burden of cancer is projected to reach INT $ 25.2 trillion from 2020 to 2050 [15], underscoring its impact not only on health but also on economic sustainability. In Japan, cancer remains one of the most pressing societal challenges and has been the leading cause of death since 1981 [16]. It accounts for the second-highest overall medical expenditure and represents the leading medical cost among adults aged 64 years or younger [17]. Hematological malignancies, such as leukemia and lymphoma, contribute to over 10% of total cancer-related medical costs. Furthermore, insurer data indicate that these malignancies constitute a significant proportion of the top 100 high-cost medical claims [18]. As the aging population continues to grow, the incidence [19,20] and mortality [21] associated with hematological malignancies continue to increase. Therefore, increased attention to these cancers is critical to maintaining a sustainable healthcare system and society.

Especially, cancer mortality is one of the most robust and key indices, as it reflects not only the incidence of cancer but also biological and clinical outcomes, as well as social factors such as access to medical and health care. Because mortality can be influenced by such social factors, it is important to focus on this outcome. Furthermore, according to the Ministry of Health, Labour, and Welfare, cancer mortality has been set as a key indicator ever since the establishment of the Basic Plan to Promote Cancer Control Programs [22–25]. Moreover, in the Basic Plan to Promote Cancer Control Programs (Phase 4), poor-prognosis cancer mortality is included as one of the indicators, although no explicit definition is provided in the plan. In practice, however, a 5-year survival rate of less than 50% [25–27] is sometimes used as a criterion for defining poor-prognosis cancers, which include leukemia and multiple myeloma [28–30]. Given these considerations, focusing on cancer mortality remains essential for understanding and addressing the overall cancer burden.

According to a pooled analysis of 20 prospective studies [31], a higher body mass index (BMI) was significantly associated with increased mortality from multiple myeloma (hazard ratio (HR): 1.09, 95% confidence interval (CI): 1.03–1.16); however, all studies included in that analysis were conducted in Western countries. With

regard to lymphoma, non-Hodgkin lymphoma mortality was significantly elevated per 5 kg/m² increase in BMI (relative risk (RR): 1.04 (1.04–1.26)) based on findings from a meta-analysis [32], although the included studies were predominantly conducted in Western populations. Similarly, in the case of leukemia, a significantly higher mortality risk was observed in individuals with obesity (RR: 1.29 (1.11–1.49)), as demonstrated in a meta-analysis of six studies [33].

In contrast to findings from Western populations, where numerous studies have demonstrated a significant association between obesity and mortality from hematological malignancies, evidence from Asian populations remains limited. A pooled analysis of over 420,000 participants from the Asia-Pacific region found no clear association, possibly due to a short follow-up period [34]. Conversely, another pooled study comprising over 800,000 Asian participants reported a significant association between BMI and multiple myeloma mortality [35]. In the Japanese population, one study conducted by the Japan Collaborative Cohort Study (the JACC study) Group examined the association between obesity and hematological malignancies [36]. However, the follow-up period was not completed at the time of the study, and the analysis was restricted to multiple myeloma. Therefore, updated evidence is needed to clarify the association between obesity and mortality due to multiple myeloma and other hematological malignancies in the Japanese population. Thus, this study aimed to identify the relationship between obesity and mortality due to hematological malignancies in a Japanese population.

## Materials and methods

### Study design

Data for this study were obtained from the JACC study, the details of which have been described previously [37]. The JACC Study was conducted from 1988 to 1990 across 45 areas in Japan and enrolled 110,585 people aged 40–79 years who completed a self-administered questionnaire at baseline. Follow-up was carried out until the end of 2009. Individuals with missing data on height or weight (n = 5,840), cancer history at baseline (n = 1,276), or a follow-up duration of <5 years (n = 6,396) were excluded. Consequently, 97,073 individuals were included in the final analysis (Fig 1).

Individual informed consent before participation in the study was obtained in 36 of the 45 study areas, with written informed consent in 35 areas and oral consent in 1 area. In the remaining 9 areas, group consent was obtained from the respective area leaders. This study was approved by the Ethics Committee of the Graduate School of Medicine at Hokkaido University (14–044).

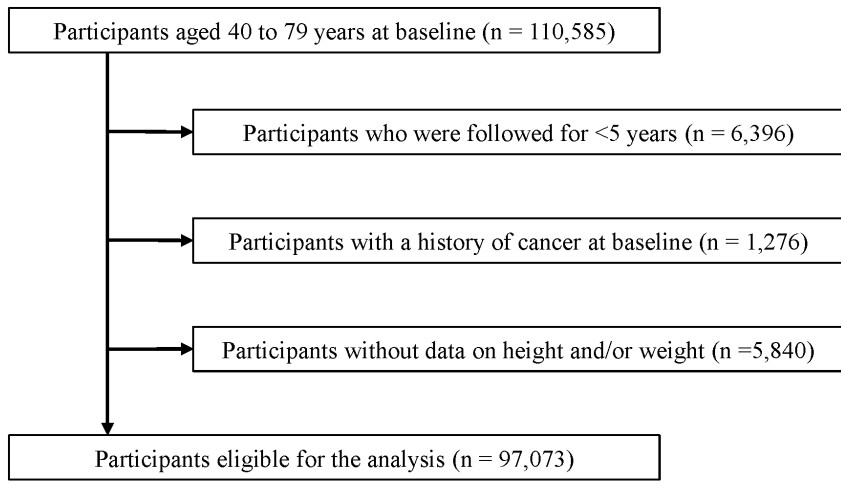

**Fig 1. Flowchart of the participants.**

## Outcomes

Mortality and relocation dates for study participants were obtained from the population registry of each study area. In addition, the causes of death were confirmed using death certificates and coded according to the International Classification of Diseases, 10th Revision. Deaths coded from C81 to 96 were defined as mortality from hematological malignancies. To further analyze specific hematological malignancies, the following categories were examined: lymphoma (C81–C86), multiple myeloma (C90), leukemia (C91-C95), and myeloid leukemia (C92).

## Exposure

BMI (kg/m$^2$) was calculated using self-reported height and weight. The participants were categorized into four BMI groups: underweight (BMI < 18.5), normal-weight (18.5 ≤ BMI < 25), overweight (25 ≤ BMI < 30), and obese (30 ≤ BMI).

## Covariates

Baseline covariates were obtained from the self-administered questionnaire and included the following variables: age, sex, childhood living area (large cities, other cities, rural areas/islands or remote areas, and others), educational attainment (<10 years, < 13 years, < 16 years, and ≥16 years), employment status at baseline (employed/part-time, self-employed, homemaker, unemployed, and others), drinking status (current drinker, former drinker, and non-drinker), smoking status (current smoker, former smoker, and non-smoker), weekly exercise habits (≥5 hours, 3–4 hours, 1–2 hours, and minimal), and daily walking habits (≥1 hour, 30 minutes to 1 hour, approximately 30 minutes, and minimal). The childhood living area was assessed using the question, "Which type of the city or area did you live in the longest until you graduated elementary school?." Educational attainment was evaluated using the question, "How old were you when you last attended school?." Given that children in Japan typically start school at the age of 6 years, the duration of education was calculated by subtracting six from the reported age and then categorized into five groups accordingly.

## Statistical analysis

Participant characteristics were expressed as numbers and percentages for categorical values. Cox proportional hazards models were employed to estimate HRs and 95% CIs for the association between BMI and mortality from hematological malignancies. For cause-specific analyses, deaths from causes other than the outcome of interest were treated as censored at the time of death. No covariates were included in Model 1. In Model 2, adjustments were made for age and sex. Model 3 further included childhood living area, educational attainment, employment status at baseline, drinking status, smoking status, weekly exercise habits, and daily walking habits. To address missing data in covariates, multiple imputation was conducted. A total of 20 imputed datasets were aggregated, and the results were pooled using Rubin's rule. A two-sided p-value of <0.05 was considered significant, and 95% CIs were used for interval estimation. All statistical analyses were performed using the R software (version 4.5.0) with the Survival package (version 3.8.3) and the Mice package (version 3.17.0).

## Results

### Participant characteristics

The baseline characteristics, including sex, age, and other demographic variables, of the 97,073 participants (40,479 men (41.7%) and 56,594 women (58.3%)) are presented in Table 1. A total of 1,671 participants (1.7%) were classified as obese. The majority of participants were of normal weight (n = 70,979 (73.1%)), followed by overweight (n = 18,976 (19.5%)) and underweight (n = 5,447 (5.6%)). Participants with obesity were more likely to be women, had a lower educational attainment, were homemakers, or were unemployed. Additionally, they were more frequently non-drinkers and

**Table 1. Participant characteristics.**

| | Overall | Underweight | Normal-weight | Overweight | Obese |
|---|---|---|---|---|---|
| | | (BMI < 18.5) | (18.5 ≤ BMI < 25) | (25 ≤ BMI < 30) | (30 ≤ BMI) |
| | 97,073 | 5,447 | 70,979 | 18,976 | 1,671 |
| BMI (kg/m²), mean (SD) | 22.86 (3.51) | 17.45 (0.97) | 22.03 (1.68) | 26.63 (1.25) | 32.99 (13.66) |
| Sex = men, n (%) | 40,479 (41.70) | 2,044 (37.53) | 30,768 (43.35) | 7,221 (38.05) | 446 (26.69) |
| Age (years), mean (SD) | 57.03 (9.90) | 61.20 (10.64) | 56.85 (9.93) | 56.52 (9.34) | 56.84 (9.56) |
| Age (years), n (%) | | | | | |
| 40s | 25,450 (26.22) | 978 (17.95) | 19,178 (27.02) | 4,872 (25.67) | 422 (25.25) |
| 50s | 30,926 (31.86) | 1,163 (21.35) | 22,485 (31.68) | 6,705 (35.33) | 573 (34.29) |
| 60s | 29,220 (30.10) | 1,921 (35.27) | 21,115 (29.75) | 5,682 (29.94) | 502 (30.04) |
| 70s | 11,477 (11.82) | 1,385 (25.43) | 8,201 (11.55) | 1,717 (9.05) | 174 (10.41) |
| Living area of childhood, n (%) | | | | | |
| Large cities | 13,543 (13.95) | 840 (15.42) | 9,919 (13.97) | 2,532 (13.34) | 252 (15.08) |
| Other cities | 4,840 (4.99) | 260 (4.77) | 3,570 (5.03) | 921 (4.85) | 89 (5.33) |
| Rural areas/Islands or remote areas | 55,857 (57.54) | 3,003 (55.13) | 40,810 (57.50) | 11,097 (58.48) | 947 (56.67) |
| Others | 3,099 (3.19) | 188 (3.45) | 2,225 (3.13) | 633 (3.34) | 53 (3.17) |
| Educational attainment, n (%) | | | | | |
| <10 years | 27,852 (28.69) | 1,692 (31.06) | 19,993 (28.17) | 5,590 (29.46) | 577 (34.53) |
| <13 years | 35,657 (36.73) | 1,900 (34.88) | 26,556 (37.41) | 6,701 (35.31) | 500 (29.92) |
| <16 years | 7148 (7.36) | 434 (7.97) | 5378 (7.58) | 1261 (6.65) | 75 (4.49) |
| ≥16years | 2411 (2.48) | 111 (2.04) | 1833 (2.58) | 445 (2.35) | 22 (1.32) |
| Current job at the baseline, n (%) | | | | | |
| Employed/Part-time | 25,796 (26.57) | 994 (18.25) | 19,680 (27.73) | 4,787 (25.23) | 335 (20.05) |
| Self-employed | 22,155 (22.82) | 1,066 (19.57) | 16,291 (22.95) | 4,445 (23.42) | 353 (21.13) |
| Homemaker | 14,953 (15.40) | 906 (16.63) | 10,597 (14.93) | 3,122 (16.45) | 328 (19.63) |
| Unemployed | 15,061 (15.52) | 1,378 (25.30) | 10,655 (15.01) | 2,738 (14.43) | 290 (17.35) |
| Others | 4,394 (4.53) | 317 (5.82) | 3,269 (4.61) | 750 (3.95) | 58 (3.47) |
| Drinking status, n (%) | | | | | |
| Current drinker | 42,542 (43.82) | 1,942 (35.65) | 32,105 (45.23) | 7,953 (41.91) | 542 (32.44) |
| Former drinker | 2,946 (3.03) | 240 (4.41) | 2,109 (2.97) | 536 (2.82) | 61 (3.65) |
| Non-drinker | 45,412 (46.78) | 2,854 (52.40) | 32,399 (45.65) | 9,235 (48.67) | 924 (55.30) |
| Smoking status, n (%) | | | | | |
| Current smoker | 23,299 (24.00) | 1,430 (26.25) | 17,818 (25.10) | 3,773 (19.88) | 278 (16.64) |
| Former smoker | 10,895 (11.22) | 492 (9.03) | 8,030 (11.31) | 2,222 (11.71) | 151 (9.04) |
| Non-smoker | 54,321 (55.96) | 2,988 (54.86) | 39,030 (54.99) | 11,241 (59.24) | 1,062 (63.55) |
| Exercise habits of the week, n (%) | | | | | |
| ≥5 hours | 4,443 (4.58) | 268 (4.92) | 3,306 (4.66) | 803 (4.23) | 66 (3.95) |
| 3–4 hours | 4,910 (5.06) | 251 (4.61) | 3,653 (5.15) | 930 (4.90) | 76 (4.55) |
| 1–2 hours | 11,944 (12.30) | 582 (10.68) | 8,909 (12.55) | 2,293 (12.08) | 160 (9.58) |
| Minimal | 57,251 (58.98) | 3,227 (59.24) | 41,527 (58.51) | 11,414 (60.15) | 1,083 (64.81) |
| Walking habits of the day, n (%) | | | | | |
| ≥1 hour | 37,852 (38.99) | 2,184 (40.10) | 28,290 (39.86) | 6,855 (36.12) | 523 (31.30) |
| 30 min to 1 hour | 14,937 (15.39) | 847 (15.55) | 11,011 (15.51) | 2,829 (14.91) | 250 (14.96) |
| Around 30 min | 13,230 (13.63) | 783 (14.37) | 9,524 (13.42) | 2,695 (14.20) | 228 (13.64) |
| Minimal | 8,405 (8.66) | 456 (8.37) | 5,907 (8.32) | 1,837 (9.68) | 205 (12.27) |

For each variable, percentages may not total 100% due to missing data.

non-smokers and reported lower levels of physical activity and walking compared with individuals in other BMI categories. Conversely, those in the underweight category tended to be older, unemployed, and non-drinkers.

## BMI and cause-specific mortality from hematological malignancies

Over a mean follow-up duration of 17 years, a total of 479 participants died from hematological malignancies were recorded. These included 200 participants who died from lymphoma (41.7%), 107 from multiple myeloma (22.3%), and 166 from leukemia (34.5%), of whom 106 died due to myeloid leukemia (63.9% of leukemia deaths).

Fig 2 illustrates the HRs estimated in each model for the associations between BMI categories and mortality from hematological malignancies, with the normal-weight group serving as the reference. In Model 3, obesity was significantly associated with a higher risk of mortality from all hematological malignancies combined (HR: 1.78, 95% CI: 1.02–3.11). In disease-specific subgroup analyses, obesity was significantly associated with higher mortality from multiple myeloma (HR: 2.75, 95% CI: 1.09–6.94), leukemia (HR: 2.47, 95% CI: 1.07–5.69), and particularly myeloid leukemia (HR: 3.89, 95% CI: 1.66–9.11) in Model 3. By contrast, no significant association was observed between BMI and lymphoma-related mortality in any of the models. The HRs for each covariate included in the multivariable models are presented in S1 Table.

## Discussion

A significant association was found between obesity and mortality due to hematological malignancies in the Japanese population. Moreover, obesity was associated with increased mortality from multiple myeloma and leukemia, particularly myeloid leukemia.

The results of the present study are consistent with those of previous studies on the association between obesity and hematological malignancies. In the case of multiple myeloma, a meta-analysis of 13 prospective studies reported that the RR of multiple myeloma mortality was significantly higher among individuals with obesity compared with those of normal weight (RR: 1.54, 95% CI: 1.35–1.76) [38]. Similarly, a pooled analysis of 20 studies indicated that a higher BMI was associated with increased mortality from multiple myeloma [31]. With regard to leukemia, a meta-analysis of six studies demonstrated a significant association between obesity and leukemia-related mortality [33]. In addition, a cohort study conducted among Taiwanese adults also identified a significant association between obesity and mortality from leukemia (HR: 2.30, 95% CI: 1.16–4.58) [39].

Although the mechanisms underlying the association between obesity and hematological malignancies have not been fully established, several potential pathways have been proposed. First, with regard to disease development, one plausible mechanism involves clonal hematopoiesis—a condition associated with an increased risk of hematological malignancies. Evidence indicates that clonal hematopoiesis of indeterminate potential (CHIP) is more frequently observed in individuals with obesity. A study investigating the association between lifestyle factors and CHIP among women reported that individuals with normal weight and overweight had lower ORs for the presence of CHIP compared with those with obesity [40]. Thus, the risk of the presence of CHIP is the highest for those with obesity. Furthermore, another study demonstrated that individuals with clonal hematopoiesis had a significantly higher HR for developing hematologic cancers (HR: 12.9, 95% CI: 5.8–28.7) compared with those without detectable somatic mutations [41]. Additionally, an analysis using data from the Biobank identified a strong association between clonal hematopoiesis and the incidence of myeloid malignancies [42].

A second possible mechanism underlying disease development involves bone marrow adipose tissue (BMAT). The vertebrae remain primary sites of hematopoiesis throughout adulthood; however, visceral fat accumulation has been associated with increased vertebral BMAT [43]. BMAT expansion compromises normal hematopoiesis by reducing the number of hematopoietic stem cells (HSCs) [44], which may, in turn, contribute to the development of hematologic malignancies [45].

A third proposed mechanism involves adipokines—bioactive substances secreted by adipose tissue that regulate inflammation, immune responses, and endocrine functions. Among these, adiponectin, which is typically found at reduced

| | Time at risk (Person-years) | Mortality (%) | Model 1 HR (95% CI) | Model 2 HR (95% CI) | Model 3 HR (95% CI) | |
|---|---|---|---|---|---|---|
| **All hematological malignancies (C81-C96)** | | | | | | |
| Underweight (BMI <18.5) | 84,382 | 30 (0.55) | 1.32 (0.91, 1.91) | 1.15 (0.79, 1.68) | 1.14 (0.78, 1.66) | |
| Normal-weight (18.5≤BMI<25) | 1,211,628 | 351 (0.49) | 1(ref) | 1(ref) | 1(ref) | |
| Overweight (25≤BMI<30) | 327,241 | 85 (0.45) | 0.89 (0.70, 1.13) | 0.93 (0.74, 1.18) | 0.94 (0.74, 1.20) | |
| Obese(30≤BMI) | 28,133 | 13 (0.78) | 1.61 (0.93, 2.81) | 1.77 (1.02, 3.09) | 1.78 (1.02, 3.11) | |
| **Lymphoma (C81-C86)** | | | | | | |
| Underweight (BMI <18.5) | 84,382 | 12 (0.22) | 1.21 (0.67, 2.18) | 1.06 (0.59, 1.92) | 1.04 (0.57, 1.88) | |
| Normal-weight (18.5≤BMI<25) | 1,211,628 | 152 (0.21) | 1(ref) | 1(ref) | 1(ref) | |
| Overweight (25≤BMI<30) | 327,241 | 34 (0.18) | 0.82 (0.57, 1.19) | 0.87 (0.60, 1.27) | 0.90 (0.62, 1.32) | |
| Obese(30≤BMI) | 28,133 | 2 (0.12) | 0.57 (0.14, 2.31) | 0.65 (0.16, 2.63) | 0.67 (0.16, 2.73) | |
| **Multiple myeloma (C90)** | | | | | | |
| Underweight (BMI <18.5) | 84,382 | 5 (0.09) | 0.96 (0.39, 2.38) | 0.79 (0.32, 1.96) | 0.76 (0.30, 1.92) | |
| Normal-weight (18.5≤BMI<25) | 1,211,628 | 80 (0.11) | 1(ref) | 1(ref) | 1(ref) | |
| Overweight (25≤BMI<30) | 327,241 | 17 (0.09) | 0.78 (0.46, 1.32) | 0.81 (0.48, 1.37) | 0.80 (0.47, 1.36) | |
| Obese(30≤BMI) | 28,133 | 5 (0.30) | 2.71 (1.10, 6.69) | 2.84 (1.15, 7.02) | 2.75 (1.09, 6.94) | |
| **Leukemia (C91-C95)** | | | | | | |
| Underweight (BMI <18.5) | 84,382 | 13 (0.24) | 1.75 (0.98, 3.10) | 1.60 (0.90, 2.85) | 1.61 (0.90, 2.89) | |
| Normal-weight (18.5≤BMI<25) | 1,211,628 | 115 (0.16) | 1(ref) | 1(ref) | 1(ref) | |
| Overweight (25≤BMI<30) | 327,241 | 32 (0.17) | 1.02 (0.69, 1.52) | 1.06 (0.72, 1.57) | 1.06 (0.72, 1.58) | |
| Obese(30≤BMI) | 28,133 | 6 (0.36) | 2.28 (1.00, 5.17) | 2.47 (1.08, 5.62) | 2.47 (1.07, 5.69) | |
| **Myeloid leukemia (C92)** | | | | | | |
| Underweight (BMI <18.5) | 84,382 | 6 (0.11) | 1.19 (0.52, 2.73) | 1.08 (0.47, 2.50) | 1.11 (0.48, 2.61) | |
| Normal-weight (18.5≤BMI<25) | 1,211,628 | 78 (0.11) | 1(ref) | 1(ref) | 1(ref) | |
| Overweight (25≤BMI<30) | 327,241 | 16 (0.08) | 0.76 (0.44, 1.30) | 0.80 (0.47, 1.38) | 0.80 (0.46, 1.38) | |
| Obese(30≤BMI) | 28,133 | 6 (0.36) | 3.37 (1.47, 7.74) | 3.87 (1.68, 8.90) | 3.89 (1.66, 9.11) | |

Model 3
HR (95% CI)

**Fig 2. Hazard ratios for the associations between BMI categories and mortality from hematological malignancy.** Model 1 was the crude model. Model 2 was adjusted for sex and age. Model 3 was further adjusted for childhood living area, educational attainment, employment status at baseline, drinking status, smoking status, weekly exercise habits, and daily walking habits.

levels in individuals with obesity [46,47], plays a critical role in regulating inflammatory cytokines and maintaining hematopoietic stem cell self-renewal and quiescence [48]. Moreover, certain adipokines induce apoptosis in multiple myeloma cells [49].

In addition, chronic inflammation caused by obesity [50] has been implicated as a potential contributor to the pathogenesis of hematopoietic diseases [51]. Consequently, although the precise biological mechanisms still need to be elucidated, obesity may lead to the development of multiple myeloma or leukemia.

Beyond disease development, obesity may also negatively affect treatment outcomes. For instance, patients with acute myeloid leukemia who were classified as obese demonstrated poorer overall survival compared with normal-weight [52] or non-obese patients [53]. Similarly, patients classified as overweight or obese had lower overall survival and leukemia-free survival following allogeneic hematopoietic stem cell transplantation (allo-HSCT) [54]. In patients with multiple myeloma, a higher BMI at the time of autologous hematopoietic stem cell transplantation (auto-HSCT) was negatively associated with overall survival [55]. Additionally, a lower rate of progression-free survival after auto-HSCT was observed among patients with obesity [56]. Taken together, these findings suggest that obesity may lead to worse outcomes in patients with hematological malignancies.

In the present study, no significant association was observed between obesity and lymphoma-related mortality. One possible explanation for this finding is the heterogeneity inherent in lymphomas. Given the wide range of lymphoma subtypes, the potential effect of obesity on individual subtypes may not have been detectable in the aggregated analysis. A previous meta-analysis [57] demonstrated that the RR associated with obesity and non-Hodgkin's lymphoma varied across subgroups. Moreover, previous studies reported inconsistent results regarding the relationship between obesity and lymphoma-related mortality [34,39,58,59]. Furthermore, analysis of data from 2000–2008 revealed notable differences in the 5-year survival rates, with lymphoma showing 54.6–65.5%, compared with 29.0–36.4% for multiple myeloma and 32.1–39.2% for leukemia [28–30]. Therefore, this may partly explain why a significant association for lymphoma mortality was not observed in the present study.

However, this study has some limitations. First, self-reported data were used to assess height and weight, which may have introduced misclassification bias. Previous research has indicated a tendency for individuals to underreport weight and overreport height [60], potentially leading to an underestimation of obesity prevalence. Second, as the follow-up period ended in 2009, more recent advancements in treatment—particularly in chemotherapy and other therapeutic modalities—could not be accounted for. Given the rapid evolution of treatment strategies in recent years, outcomes in the current population may differ from those observed in this study. Third, this study did not assess the magnitude of the associations with respect to incidence or prognosis; thus, careful interpretation of the results is necessary. Fourth, due to using data from the death certificates, misclassification might occur for the death causes. However, a previous study found that the concordance between autopsy results and the cause of death on the death certificates was relatively high for cancer (81%). Therefore, it will not significantly affect our results [61]. Additionally, body weight and height were measured only at baseline, and no information on weight changes during follow-up was available. Therefore, we could not assess the potential impact of weight change over time on the association between BMI and mortality. Nevertheless, this may not critically affect our results, as previous Japanese epidemiological study have shown that the body weight of middle-aged and older adults tends to be relatively stable [62]. Moreover, participants with less than five years of follow-up were excluded to minimize reverse causation due to preclinical disease, residual bias from unmeasured weight changes cannot be completely ruled out. Finally, the stratified analysis, which would have allowed us to evaluate the influence of important covariates such as sex and smoking, was not feasible due to the insufficient number of mortalities.

This study possesses several strengths. First, the long follow-up duration was likely sufficient to evaluate the long-term outcomes. Second, as the participants were recruited from multiple regions across Japan, the findings may be representative of the Japanese population.

## Conclusion

This study examined the association between obesity and mortality from hematological malignancies in the Japanese population, identifying significant associations with mortality due to leukemia and multiple myeloma. These findings are consistent with those reported in previous studies conducted in the Western population. These findings suggest that obesity may be an important modifiable risk factor for certain hematological malignancies. While the present study focused on mortality as the outcome, future studies should investigate the association between BMI and cancer incidence in the Japanese population to provide a more comprehensive understanding of their relationship.

## Supporting information

**S1 Table. HRs and 95% CIs for all covariates for model 2 and model 3.**
(PDF)

## Acknowledgments

We sincerely appreciate all participants and staff members for their invaluable contributions and dedicated involvement in the conduct of this study.

## Author contributions

**Conceptualization:** Hana Wakasa.

**Data curation:** Hana Wakasa, Satoshi Sunohara, Takashi Kimura.

**Formal analysis:** Satoshi Sunohara.

**Funding acquisition:** Takashi Kimura, Akiko Tamakoshi.

**Investigation:** Akiko Tamakoshi.

**Methodology:** Hana Wakasa, Satoshi Sunohara.

**Resources:** Akiko Tamakoshi.

**Supervision:** Akiko Tamakoshi.

**Validation:** Hana Wakasa.

**Visualization:** Satoshi Sunohara.

**Writing – original draft:** Hana Wakasa, Satoshi Sunohara.

**Writing – review & editing:** Hana Wakasa, Satoshi Sunohara, Takashi Kimura, Takaya Ichikawa, Akiko Tamakoshi.

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
