## [Decision Letter · Decision Letter 0]

7 Aug 2025

Dear Dr. Tamakoshi,

Thank you for submitting your manuscript to PLOS ONE. After careful consideration, we feel that it has merit but does not fully meet PLOS ONE’s publication criteria as it currently stands. Therefore, we invite you to submit a revised version of the manuscript that addresses in detail all the points raised during the review process by both Reviewers.

We look forward to receiving your revised manuscript.

Kind regards,

Francesco Bertolini, MD, PhD

Academic Editor

PLOS ONE

 [AT received funding from Grants-in-Aid for Scientific Research from the Ministry of Education, Culture, Sports, Science and Technology of Japan (MEXT) (Monbusho); Grants-in-Aid for Scientific Research on Priority Areas of Cancer; Grants-in-Aid for Scientific Research on Priority Areas of Cancer Epidemiology from MEXT (MonbuKagaku-sho) (Nos. 61010076, 62010074, 63010074, 1010068, 2151065, 3151064, 4151063, 5151069, 6279102, 11181101, 17015022, 18014011, 20014026, 20390156, and 26293138); Japan Society for the Promotion of Science, JSPS KAKENHI Grant Number JP 16H06277 (CoBiA) , and 22H04923;  grant–in–aid from the Ministry of Health, Labor and Welfare, Health and Labor Sciences research grants, Japan (Research on Health Services: H17–Kenkou–007; Comprehensive Research on Cardiovascular Disease and Life–Related Disease: H18–Junkankitou [Seishuu]–Ippan–012; H19–Junkankitou [Seishuu]–Ippan–012; H20–Junkankitou [Seishuu]–Ippan–013; H23–Junkankitou [Seishuu]–Ippan–005; H26-Junkankitou [Seisaku]-Ippan-001; H29–Junkankitou–Ippan–003; 20FA1002; 23FA1006); the National Cancer Center Research and Development Fund (27-A-4, 30-A-15, 2021-A-16, 2024-A-14)]. 

Additional Editor Comments (if provided):

Reviewers' comments:

Reviewer's Responses to Questions

**Comments to the Author**

1. Is the manuscript technically sound, and do the data support the conclusions?

Reviewer #1: Yes

Reviewer #2: No

2. Has the statistical analysis been performed appropriately and rigorously?

Reviewer #1: Yes

Reviewer #2: No

3. Have the authors made all data underlying the findings in their manuscript fully available?

Reviewer #1: Yes

Reviewer #2: Yes

4. Is the manuscript presented in an intelligible fashion and written in standard English?

Reviewer #1: Yes

Reviewer #2: Yes

Reviewer #1: This study makes a potentially important contribution to understanding obesity-related cancer mortality in Japanese adults. Overall, the manuscript is well written. However, there are several methodological and interpretational issues that should be addressed to strengthen the validity and clarity of the findings.

1. As the authors used mortality rather than incidence as the primary outcome, it is important to acknowledge that the observed associations may reflect not only disease risk but also differences in survival. For hematologic malignancies with relatively favorable prognosis, such as certain subtypes of lymphoma, associations with BMI may be attenuated or obscured when mortality is used as the endpoint. Although the discussion briefly touches upon this point, the authors should more clearly elaborate on the implications of using mortality as the outcome.

A more explicit consideration of this limitation would help contextualize the null findings for lymphoma and strengthen the overall interpretation. Furthermore, given that the JACC Study likely possesses incidence data based on previous publications, a discussion of whether the association between BMI and hematologic malignancies would differ if incidence (rather than mortality) were used as the outcome would greatly enhance the depth and value of the manuscript.

2. BMI was measured only at baseline. Please clarify whether any sensitivity analyses were conducted to account for potential weight changes over time or reverse causation, particularly given the long duration of follow-up.

3. The cause of death was determined from death certificates, which may lack diagnostic specificity, especially for subtypes of hematological malignancies. Please consider adding a discussion on the potential for misclassification of cancer subtypes.

4. Smoking is an important risk factor for hematological malignancies and is also closely associated with BMI. Therefore, conducting analyses stratified by smoking status or evaluating potential interactions between smoking and BMI would be informative.

5. Were there any differences by sex? Given possible sex differences in BMI distribution and hematologic cancer risk, stratified analyses by sex would also be valuable.

Reviewer #2: Review to the Authors in Plos one 2025 Aug

Thank you for the opportunity of reviewing the article on "Association between obesity and mortality from hematological malignancies among Japanese adults: The Japan Collaborative Cohort study. The cohort study length is astonishing and has produced many academic findings; however, there are several concerns for publication.

Major comments

1 Is this study assessing association between obesity in 1990 and mortality in later period among patients suffering from hematological malignancies? Mortality can be affected by many other factors such as renal dysfunction, diabetes, hyperlipidemia, hypertension, arteriosclerosis, etc. Without assessing these factors, the true association between obesity in about 20 years ago and mortality in later life is impossible to clarify. As the authors wrote in line 203, renal dysfunction, diabetes, and arteriosclerosis, etc, negatively affect treatment outcomes. Therefore, just assessing obesity in the past and mortality is meaningless.

Instead, the author should explore an association between obesity and incidence of hematological malignancy in later period. CHIP theory may demonstrate that obese people may have an increased incidence of hematological malignancy.

2 In Figure 1, overweight group shows a lowered risk though they are not statistically significant. CHIP and BMAT theory cannot explain the opposite trend. How do you explain?

3 In Figure 1, if the authors included covariates such as age, sex, drinking status etc in model 3, all HRs and 95% CIs should be shown. Is there any other variable associated with the mortality of hematological malignancies?

4 The average age (SD) or age group must be shown. Age has the strongest impact on mortality in general. By the way, I cannot find Table 1.

**Do you want your identity to be public for this peer review?** For information about this choice, including consent withdrawal, please see our Privacy Policy

Reviewer #1: No

Reviewer #2: No

---

## [Author Response · Author response to Decision Letter 1]

1 Sep 2025

We are most grateful to you and the reviewers for the comments on our manuscript. These have greatly helped us to improve our manuscript.　We have revised our manuscript based on the constructive suggestions of the reviewers. The follows are our responses. Please see the attached file "response to reviewers" for more information.

----

Revision Note

We are most grateful to you and the reviewers for the comments on our manuscript. These have greatly helped us to improve our manuscript.　We have revised our manuscript based on the constructive suggestions of the reviewers.

Please note, in the responses below, we have included extra citations that are used only for explaining our rationale to the reviewers and are not included in the main manuscript. Such response-only references are indicated in the format “(R1, R2……)”.

The following are our responses to the reviewer 1 and 2. Our responses are in blue, and the changes are in italics. Also, the changes are shown in red in the revised manuscript.

Reviewer 1

1. As the authors used mortality rather than incidence as the primary outcome, it is important to acknowledge that the observed associations may reflect not only disease risk but also differences in survival. For hematologic malignancies with relatively favorable prognosis, such as certain subtypes of lymphoma, associations with BMI may be attenuated or obscured when mortality is used as the endpoint. Although the discussion briefly touches upon this point, the authors should more clearly elaborate on the implications of using mortality as the outcome.

A more explicit consideration of this limitation would help contextualize the null findings for lymphoma and strengthen the overall interpretation. Furthermore, given that the JACC Study likely possesses incidence data based on previous publications, a discussion of whether the association between BMI and hematologic malignancies would differ if incidence (rather than mortality) were used as the outcome would greatly enhance the depth and value of the manuscript.

Author response

We thank the reviewer for this important and insightful comment. We appreciate the opportunity to clarify the rationale behind our choice of mortality as the primary outcome in this study.

The primary objective of our research was to investigate the association between BMI and mortality from hematological malignancies. Mortality is a critical endpoint because it reflects not only the risk of developing the disease but also the integrated biological and clinical consequences after diagnosis, including treatment response, toxicity, recurrence, and survival. From a public health perspective, reducing cancer mortality is a major goal, as emphasized in Japan’s national cancer control policy [R1]. Thus, studying mortality provides essential evidence to guide strategies aimed at improving long-term outcomes for patients and society.

We agree that, for hematologic malignancies with relatively favorable prognoses such as lymphoma, the use of mortality as an outcome may attenuate or obscure associations with BMI. In our study, the lack of a statistically significant association between BMI and lymphoma mortality may indeed be partly attributable to the higher survival rate of lymphoma compared with other hematologic malignancies. To address this point, we have now added a sentence in the Discussion noting that differences in survival rates among hematologic malignancies could have influenced the observed associations.

We acknowledge your suggestion regarding the potential value of analyzing the association between BMI and the incidence of hematological malignancies. We are aware that the JACC Study includes incidence data; however, these data are limited to certain geographic areas and thus pertain to a subset of participants different from those included in the present analysis. Due to these differences in the study population and data availability, incorporating incidence analyses within this manuscript was not feasible.

Nonetheless, we recognize the importance of this research question and plan to address it in future studies using the incidence data. We have added the statement in the Conclusion section to highlight future research directions.

In summary, we have revised the Discussion and the Conclusion section to more explicitly address the limitations and implications of using mortality as the primary outcome, particularly in relation to hematological malignancies with favorable prognosis such as certain lymphoma subtypes. We appreciate your valuable input, which has helped us to strengthen the contextual interpretation of our findings and to outline clear avenues for future investigation.

Change

(page 4)

Especially, cancer mortality is one of the most robust and key indices, as it reflects not only the incidence of cancer but also biological and clinical outcomes, as well as social factors such as access to medical and health care. Because mortality can be influenced by such social factors, it is important to focus on this outcome. Furthermore, according to the Ministry of Health, Labour, and Welfare, cancer mortality has been set as a key indicator ever since the establishment of the Basic Plan to Promote Cancer Control Programs [22–25]. Moreover, in the Basic Plan to Promote Cancer Control Programs (Phase 4), poor-prognosis cancer mortality is included as one of the indicators, although no explicit definition is provided in the plan. In practice, however, a 5-year survival rate of less than 50% [25–27] is sometimes used as a criterion for defining poor-prognosis cancers, which include leukemia and multiple myeloma [28–30]. Given these considerations, focusing on cancer mortality remains essential for understanding and addressing the overall cancer burden.

(page 18)

While the present study focused on mortality as the outcome, future studies should investigate the association between BMI and cancer incidence in the Japanese population to provide a more comprehensive understanding of their relationship.

(page 16)

Furthermore, analysis of data from 2000–2008 revealed notable differences in the 5-year survival rates, with lymphoma showing 54.6–65.5%, compared with 29.0–36.4% for multiple myeloma and 32.1–39.2% for leukemia [28-30].

2. BMI was measured only at baseline. Please clarify whether any sensitivity analyses were conducted to account for potential weight changes over time or reverse causation, particularly given the long duration of follow-up.

Author response

Thank you for your important comment.

Body weight and height were measured only at baseline, and no follow-up data on these variables were available, making sensitivity analyses accounting for weight changes over time unfeasible.

However, a study conducted among Japanese showed that the weight tends to be relatively stable in the middle to older adult population. Thus, the effect of changing weight might not be crucial on our results.

In addition, to address potential reverse causation, we have already excluded participants with less than five years of follow-up from the analysis. This approach reduces the likelihood of bias caused by weight loss or gain due to preclinical disease.

We have added a statement to the Discussion section to acknowledge the absence of longitudinal weight data as a limitation of the study.

Change

(page 17)

Additionally, body weight and height were measured only at baseline, and no information on weight changes during follow-up was available. Therefore, we could not assess the potential impact of weight change over time on the association between BMI and mortality. Nevertheless, this may not critically affect our results, as previous Japanese epidemiological study have shown that the body weight of middle-aged and older adults tends to be relatively stable [53]. Moreover, participants with less than five years of follow-up were excluded to minimize reverse causation due to preclinical disease, residual bias from unmeasured weight changes cannot be completely ruled out.

3. The cause of death was determined from death certificates, which may lack diagnostic specificity, especially for subtypes of hematological malignancies. Please consider adding a discussion on the potential for misclassification of cancer subtypes.

Author response

Thank you for your insightful feedback. We appreciate your point regarding the potential for misclassification of hematological malignancies due to the use of death certificates.

We have added a discussion to address this potential limitation. We acknowledge that the lack of diagnostic specificity in death certificates could theoretically lead to misclassification of cancer subtypes. However, we also cite a previous study that reported a high concordance rate (81%) between autopsy results and death certificates for cancer deaths. Based on this evidence, we believe that any misclassification would not significantly impact our study's main conclusions. This new addition clarifies the potential limitation while providing evidence to support the validity of our findings.

Change

(page 16)

Fourth, due to using data from the death certificates, misclassification might occur for the death causes. However, a previous study found that the concordance between autopsy results and the cause of death on the death certificates was relatively high for cancer (81%). Therefore, it will not significantly affect our results [52].

4. Smoking is an important risk factor for hematological malignancies and is also closely associated with BMI. Therefore, conducting analyses stratified by smoking status or evaluating potential interactions between smoking and BMI would be informative.

Author response

Thank you for the valuable suggestion regarding stratification by smoking status and evaluation of potential interactions between smoking and BMI.

We conducted stratified analyses by smoking status to explore this; the results showed a similar trend of increased hazard ratios for obesity. However, due to the relatively small number of deaths in some strata, the confidence intervals became wide, and in certain models, convergence issues arose. These factors made interpretation difficult. The attached figure is provided for your reference, showing the results up to Model 2. As you can see, the trend remains consistent even with the stratification.

Therefore, we decided to present the overall results while acknowledging this limitation in the Discussion section.

Change

(page 17)

Finally, the stratified analysis, which would have allowed us to evaluate the influence of important covariates such as sex and smoking, was not feasible due to the insufficient number of mortalities.

5. Were there any differences by sex? Given possible sex differences in BMI distribution and hematologic cancer risk, stratified analyses by sex would also be valuable.

Author response

Thank you for the valuable suggestion regarding stratification by sex.

As with the smoking-stratified analyses, we made a concerted effort to conduct stratified analyses by sex to explore this. The results consistently showed a similar trend of increased hazard ratios for obesity in both men and women. However, as in the smoking-stratified analyses, the relatively small number of deaths in some strata led to wide confidence intervals, and in certain models, convergence issues arose. These factors made reliable interpretation challenging. The attached figure is provided for your reference, showing the results up to Model 2. As you can see, the trend remains consistent even with the stratification.

Therefore, we decided to present the overall results while clearly acknowledging this limitation in the Discussion section.

Change

(page 17)

Finally, the stratified analysis, which would have allowed us to evaluate the influence of important covariates such as sex and smoking, was not feasible due to the insufficient number of mortalities.

Reviewer 2

1 Is this study assessing association between obesity in 1990 and mortality in later period among patients suffering from hematological malignancies? Mortality can be affected by many other factors such as renal dysfunction, diabetes, hyperlipidemia, hypertension, arteriosclerosis, etc. Without assessing these factors, the true association between obesity in about 20 years ago and mortality in later life is impossible to clarify. As the authors wrote in line 203, renal dysfunction, diabetes, and arteriosclerosis, etc, negatively affect treatment outcomes. Therefore, just assessing obesity in the past and mortality is meaningless.

Instead, the author should explore an association between obesity and incidence of hematological malignancy in later period. CHIP theory may demonstrate that obese people may have an increased incidence of hematological malignancy.

Author response

Thank you for your careful review and insightful comments. We appreciate you raising these important points, particularly the role of potential confounders and the study's scope.

Regarding your first point, we agree that various clinical factors such as renal dysfunction, diabetes, hyperlipidemia, hypertension, and arteriosclerosis can significantly influence mortality in patients with hematological malignancies. However, in our conceptual framework, these conditions can be considered as the key mediators in the causal pathway between obesity and mortality. Obesity is a well-established risk factor for these comorbidities, and these conditions, in turn, can compromise treatment tolerance and long-term survival. Adjusting for these factors would, therefore, lead to an underestimation of the total effect of obesity on mortality [R2].

Regarding the concern about using past BMI data, we have referenced previous Japanese epidemiological studies that show body weight and BMI tend to be relatively stable in the adult population. This suggests that BMI from the early 1990s is a reasonable proxy for long-term obesity exposure in our study cohort. We have added about this limitation in the discussion section.

We also respectfully disagree with the suggestion to focus solely on the incidence of hematological malignancies. Mortality is a critical endpoint because obesity’s impact extends beyond cancer incidence to treatment response, toxicity, recurrence, and overall survival. Mortality reflects the integrated biological and clinical consequences of obesity—including effects on immune function, pharmacokinetics, and comorbidities—which are highly relevant after diagnosis. Moreover, in Japan’s national cancer control policy, reduction of cancer mortality is explicitly identified as a major public health goal [R1]. Providing robust evidence on modifiable risk factors for cancer mortality is therefore essential for developing effective prevention and intervention strategies that can guide policy and practice. We have added a statement in the Conclusion section to highlight future research directions

We acknowledge that investigating the incidence of hematological malignancies is also important. However, such analyses require a more restricted subset of participants and are beyond the scope of the present study. We plan to address this topic in future research. We have added the statement in the Conclusion section to highlight future research directions.

We hope these clarifications address your concerns and underscore the importance of our focus on mortality as the primary outcome.

Change

(page 4)

Especially, cancer mortality is one of the most robust and key indices, as it reflects not only the incidence of cancer but also biological and clinical outcomes, as well as social factors such as access to medical and health care. Because mortality can be influenced by such social factors, it is important to focus on this outcome. Furthermore, according to the Ministry of Health, Labour, and Welfare, cancer mortality has been set as a key indicator ever since the establishment of the Basic Plan to Promote Cancer Control Programs [22–25]. Moreover, in the Basic Plan to Promote Cancer Control Programs (Phase 4), poor-prognosis cancer mortality is included as one of the indicators, although no explicit definition is provided in the plan. In practice, howeve

---

## [Decision Letter · Decision Letter 1]

17 Sep 2025

Dear Dr. Tamakoshi,

Thank you for submitting your manuscript to PLOS ONE. After careful consideration, we feel that it has merit but does not fully meet PLOS ONE’s publication criteria as it currently stands. Therefore, we invite you to submit a revised version of the manuscript that addresses the points raised during the review process by Reviewer #2.

We look forward to receiving your revised manuscript.

Kind regards,

Francesco Bertolini, MD, PhD

Academic Editor

PLOS ONE

Journal Requirements:

Reviewers' comments:

Reviewer's Responses to Questions

**Comments to the Author**

Reviewer #1: All comments have been addressed

Reviewer #2: (No Response)

2. Is the manuscript technically sound, and do the data support the conclusions?

Reviewer #1: Yes

Reviewer #2: Partly

3. Has the statistical analysis been performed appropriately and rigorously?

Reviewer #1: Yes

Reviewer #2: No

4. Have the authors made all data underlying the findings in their manuscript fully available?

Reviewer #1: Yes

Reviewer #2: No

5. Is the manuscript presented in an intelligible fashion and written in standard English?

Reviewer #1: Yes

Reviewer #2: Yes

Reviewer #1: The authors have appropriately addressed all of my comments, and I sincerely thank them for their careful and thoughtful revisions.

Reviewer #2: Review to the Authors in Plos one 2025 Sep

Thank you for the opportunity of reviewing the revised article on "Association between obesity and mortality from hematological malignancies among Japanese adults: The Japan Collaborative Cohort study.

Major comments

1 I understand that clinical diseases such as DM and renal dysfunction may be the key mediators in the causal pathway. Although the authors are applying the full model in model 3 in Figure 3 (traditional methods), not causal regression analysis for each dependent variable, I think that is OK. I think full adjusted model (traditional) is better than the causal regression analysis to interpret the results. Because causal analysis framework can vary largely depending on researchers, it causes large difference in OR and RR etc. So, I understand your explanation, not adjusting clinical diseases.

2 In figure 2, the authors must show all HRs with age, sex, and other variables. Although the authors explained that “presenting all HRs for all covariates could dilute the emphasis on BMI,” the journal readers do not focus on the only BMI. We want to see the strength of all risk factors the authors included. Therefore, we can interpret the results of BMI on mortality.

Besides, concealing the other HRs in manuscript can lead to salami publication. To avoid it, the authors must present all HRs in any cases. If there is a problem in volume, how about creating supplementary tables.

3 in Figure 2, as for time at risk (person-years), the numbers in 4 body weight group are all the same in all analyses (malignancies, lymphoma, MM, leukemia, and Myeloid leukemia. i.e: 84382, 1211682, 327241, 28133). Generally, mortality rate should be exclusive by diseases. For instance, what happens to an obese person who died from multiple myeloma? How was the patient handled in leukemia analysis? Was the patient obese and he/she survived in leukemia analysis?

In leukemia analysis, patients who died from lymphoma, MM, myeloid leukemia should be excluded from the beginning.

**Do you want your identity to be public for this peer review?** For information about this choice, including consent withdrawal, please see our Privacy Policy

Reviewer #1: No

Reviewer #2: No

---

## [Author Response · Author response to Decision Letter 2]

26 Sep 2025

We are most grateful to you and the reviewers for the comments on our manuscript. These have greatly helped us to improve our manuscript.　We have revised our manuscript based on the constructive suggestions of the reviewers. The follows are our responses. Please see the attached file "response to reviewers" for more

information.

----

Reviewer 1

1. The authors have appropriately addressed all of my comments, and I sincerely thank them for their careful and thoughtful revisions.

Author response

We would like to express our sincere gratitude for your thoughtful and constructive feedback throughout the review process. Your kind words and valuable suggestions have been instrumental in improving the quality of our work.

Reviewer 2

1. I understand that clinical diseases such as DM and renal dysfunction may be the key mediators in the causal pathway. Although the authors are applying the full model in model 3 in Figure 3 (traditional methods), not causal regression analysis for each dependent variable, I think that is OK. I think full adjusted model (traditional) is better than the causal regression analysis to interpret the results. Because causal analysis framework can vary largely depending on researchers, it causes large difference in OR and RR etc. So, I understand your explanation, not adjusting clinical diseases.

Author response

We sincerely appreciate your generous understanding and for your thoughtful evaluation of our approach. Your supportive comments are greatly appreciated.

2. In figure 2, the authors must show all HRs with age, sex, and other variables. Although the authors explained that “presenting all HRs for all covariates could dilute the emphasis on BMI,” the journal readers do not focus on the only BMI. We want to see the strength of all risk factors the authors included. Therefore, we can interpret the results of BMI on mortality.

Besides, concealing the other HRs in manuscript can lead to salami publication. To avoid it, the authors must present all HRs in any cases. If there is a problem in volume, how about creating supplementary tables.

Author response

We sincerely thank the reviewer for this important and constructive comment. In response, we have now included all HRs for age, sex, and other covariates in the Supplementary Table, as suggested.

During this process, we also carefully reconsidered the categorization of several covariates　(childhood living area, educational attainment, and employment status) and made minor adjustments to improve clarity and consistency. Importantly, these modifications did not have any meaningful impact on the results or on the interpretation of BMI’s association with mortality.

Change

(Page 8)

childhood living area (large cities, other cities, rural areas/islands or remote areas, and others), educational attainment (<10 years, <13 years, <16 years, and ≥16 years), employment status at baseline (employed/part-time, self-employed, homemaker, unemployed, and others),

(Page 10)

Table 1 has been revised accordingly to correspond to the changes in covariate categorization.

(Page 2)

Compared with normal-weight individuals, those classified as obese exhibited a significantly higher risk of mortality from all hematological malignancies (HR: 1.78; 95% CI: 1.02–3.11), multiple myeloma (HR: 2.75; 95% CI: 1.09–6.94), leukemia (HR: 2.47; 95% CI: 1.07–5.69), and particularly myeloid leukemia (HR: 3.89; 95% CI: 1.66–9.11).

(Page 12)

In disease-specific subgroup analyses, obesity was significantly associated with higher mortality from multiple myeloma (HR: 2.75, 95% CI: 1.09–6.94), leukemia (HR: 2.47, 95% CI: 1.07–5.69), and particularly myeloid leukemia (HR: 3.89, 95% CI: 1.66–9.11) in Model 3.

(Fig. 2)

Fig. 2 has been revised accordingly to correspond to the changes in covariate categorization.

(S1 Table)

HRs for all variables included for each model are shown in the supporting information, S1 Table. For convenience, we have included the S1 Table below.

3. in Figure 2, as for time at risk (person-years), the numbers in 4 body weight group are all the same in all analyses (malignancies, lymphoma, MM, leukemia, and Myeloid leukemia. i.e: 84382, 1211682, 327241, 28133). Generally, mortality rate should be exclusive by diseases. For instance, what happens to an obese person who died from multiple myeloma? How was the patient handled in leukemia analysis? Was the patient obese and he/she survived in leukemia analysis?

In leukemia analysis, patients who died from lymphoma, MM, myeloid leukemia should be excluded from the beginning.

Author response

We sincerely thank the reviewer for raising this important point. Generally, in disease-specific survival analyses, deaths from other causes are treated as censored observations. Accordingly, in our analyses, we applied the same approach. For example, in the multiple myeloma analysis, deaths from other hematologic malignancies such as lymphoma and leukemia, as well as deaths from non-hematologic causes, were censored, and thus, person-years were the same in all analyses.

To clarify this procedure, we have added a description to the Methods section to explicitly state how other causes of death were handled in each disease-specific analysis.

Change

(Page 9)

For cause-specific analyses, deaths from causes other than the outcome of interest were treated as censored at the time of death.

---

## [Decision Letter · Decision Letter 2]

15 Oct 2025

Association between obesity and mortality from hematological malignancies among Japanese adults: The Japan Collaborative Cohort study

PONE-D-25-35537R2

Dear Dr. Tamakoshi,

We’re pleased to inform you that your manuscript has been judged scientifically suitable for publication and will be formally accepted for publication once it meets all outstanding technical requirements.

Kind regards,

Francesco Bertolini, MD, PhD

Academic Editor

PLOS ONE

Additional Editor Comments (optional):

Reviewers' comments:

Reviewer's Responses to Questions

**Comments to the Author**

Reviewer #2: All comments have been addressed

2. Is the manuscript technically sound, and do the data support the conclusions?

Reviewer #2: Yes

3. Has the statistical analysis been performed appropriately and rigorously?

Reviewer #2: Yes

4. Have the authors made all data underlying the findings in their manuscript fully available?

Reviewer #2: No

5. Is the manuscript presented in an intelligible fashion and written in standard English?

Reviewer #2: Yes

Reviewer #2: The authors have addressed all of my concerns, and I appreciate their careful and thoughtful revisions.

**Do you want your identity to be public for this peer review?** For information about this choice, including consent withdrawal, please see our Privacy Policy

Reviewer #2: No

---

## [Editor Report · Acceptance letter]

PONE-D-25-35537R2

PLOS ONE

Dear Dr. Tamakoshi,

I'm pleased to inform you that your manuscript has been deemed suitable for publication in PLOS ONE. Congratulations! Your manuscript is now being handed over to our production team.

Kind regards,

on behalf of

Dr. Francesco Bertolini

Academic Editor

PLOS ONE